REGISTERED REPORT PROTOCOL

# Rationale and validation of a novel mobile application probing motor inhibition: Proof of concept of CALM-IT

**Elise M. Cardinale**[1]*, **Reut Naim**[1], **Simone P. Haller**[1], **Ramaris German**[1], **Christian Botz-Zapp**[1], **Jessica Bezek**[1], **David C. Jangraw**[2], **Melissa A. Brotman**[1]

**1** Emotion & Development Branch, National Institute of Mental Health, Bethesda, MD, United States of America, **2** Department of Electrical and Biomedical Engineering, University of Vermont, Burlington, VT, United States of America

* elise.cardinale@nih.gov

## Abstract

Identification of behavioral mechanisms underlying psychopathology is essential for the development of novel targeted therapeutics. However, this work relies on rigorous, time-intensive, clinic-based laboratory research, making it difficult to translate research paradigms into tools that can be used by clinicians in the community. The broad adoption of smartphone technology provides a promising opportunity to bridge the gap between the mechanisms identified in the laboratory and the clinical interventions targeting them in the community. The goal of the current study is to develop a developmentally appropriate, engaging, novel mobile application called CALM-IT that probes a narrow biologically informed process, inhibitory control. We aim to leverage the rigorous and robust methods traditionally used in laboratory settings to validate this novel mechanism-driven but easily disseminatable tool that can be used by clinicians to probe inhibitory control in the community. The development of CALM-IT has significant implications for the ability to screen for inhibitory control deficits in the community by both clinicians and researchers. By facilitating assessment of inhibitory control outside of the laboratory setting, researchers could have access to larger and more diverse samples. Additionally, in the clinical setting, CALM-IT represents a novel clinical screening measure that could be used to determine personalized courses of treatment based on the presence of inhibitory control deficits.

## Introduction

Precise identification of behavioral mechanisms underlying psychopathology is essential to the assessment and development of novel, targeted therapeutics [1–4]. Despite efforts for clinicians to make use of research in their practice [5,6], challenges persist in the dissemination of evidence-based tools to directly probe mechanisms underlying psychopathology [7–10]. Traditionally, identification of behavioral mechanisms underlying psychopathology has relied on rigorous, time-intensive, clinic-based laboratory research [11], tools that are largely inaccessible to the clinical community. Advances in technology have resulted in promising

**Data Availability Statement:** All relevant data from this study will be made available upon study completion.

**Funding:** All work supported by the NIMH Intramural Research Program.

developments in more easily disseminatable and accessible platforms, including the use of mobile technology to measure clinical symptoms during daily life [12]. However, the majority of this work does not directly assess precise underlying behavioral mechanisms, such as inhibitory control. Furthermore, little work has focused on efficacy and feasibility of mobile applications as assessments of mechanisms underlying psychopathology specifically in pediatric samples. The current study introduces a mobile application, "CALM-IT," an engaging, patient and clinician user-friendly, accessible and disseminatable cognitive measure that probes inhibitory control, a well-validated behavioral construct that has been posited as one critical mechanism underlying childhood psychopathology broadly [10,13,14]. In developing CALM-IT, we aim to flexibly deploy the rigorous and robust methods traditionally used in laboratory settings to validate a mechanism-driven but easily disseminatable tool that can be used by clinicians to probe inhibitory control in the community.

Inhibitory control a well-defined and brain-related construct capturing an individual's ability to resist or modulate impulses and suppress prepotent or automatic behavioral responses [15]. Inhibitory control is a deeply studied construct across species [16–18], developmental periods [19–21], and research modalities [22–25]. This rich body of literature has led to the development of a number of canonical laboratory-based tasks. It has also led to the identification of an evolutionarily-conserved underlying neural circuitry involving the ventrolateral, ventromedial, and dorsolateral prefrontal cortices; the orbitofrontal cortex; the inferior frontal gyrus; and the dorsal, posterior, anterior cingulate cortices [16,20,26,27]. As such, researchers are well equipped with neuroscience-informed tools to study inhibitory control within the laboratory setting through the employment of standardized paradigms. However, these tasks tend to be time intensive, repetitive, and expensive, making them difficult to effectively deploy within in a community setting. For example, some tasks rely on eye-tracking technology [28] that requires equipment and specific environmental controls (i.e., the luminance in the room and participant head position relative to the presented stimuli) while others involve large numbers of repetitive trials presenting single simplistic stimuli one at a time (i.e., letters or shapes) that require long periods of sustained attention [29–32].

Inhibitory control also has broad clinical relevance. Research indicates that youth with psychopathology, including mood and attentional difficulties, have exhibit difficulty inhibiting responses, and exhibit aberrant neural activity when attempting to inhibit a motor response [7–10]. Emotion dysregulation broadly is posited to reflect compromised top-down control [33,34] resulting from dysfunction of the ventrolateral and dorsal lateral prefrontal cortices regulation of subcortical regions involved in the processing of emotion of lower level emotional processes [35,36]. These findings are consistent with the behavior clinicians observe in youth with severe irritability and attentional difficulties, as they often fail to inhibit a response, which can manifest clinically as temper outbursts, increased motor activity, and problems engaging in goal-directed actions [37]. While literature examining treatment of youth psychopathology, such as attention deficit hyperactivity disorder (ADHD) and irritability, using interventions specifically targeting inhibitory control is limited, previous work shows a reduction of mood symptoms, including anxiety, following simulant medication treatment for co-occurring attentional [38,39]. These findings provide some evidence suggesting that inhibitory control may reflect a key behavioral mechanism underlying the emergence of mood symptoms and treatment response. The proposed work could be of particular importance in relation to anxiety, irritability, and ADHD symptoms as these symptoms are common, co-occurring symptoms that are present across a wide range of childhood psychopathology [37,40–42], and are implicated in the later development of mood disorders including depression [43–46]. Furthermore, aberrant cognitive control more broadly has been implicated as potential mechanism associated with the presentation of anxiety [47–49],

irritability [8,50,51], and ADHD [19,52] symptoms. However, any innovations targeting inhibitory control will need to be both clinically valuable and accessible if they are to be embraced by the broader community.

There is a profound need for mental health treatment research efforts to probe underlying mechanisms to inform diagnosis and treatment decisions [1–4]. Clinical researchers are well poised to harness new technologies to increases access, enhance developmental approaches, and assess transdiagnostic behavioral mechanisms underlying psychopathology [53,54]. Efforts to translate this work to clinical settings is crucial as providing clinicians with tools to directly probe behavioral mechanisms could lead towards increased precision medicine, patient engagement, and cost-effectiveness [55–57]. However, application of these innovations in clinical settings is lagging [53,54]. Few clinical approaches attempt to directly probe the aberrant brain-based behavioral mechanisms that have been linked to psychopathology [58] while also providing practical use in clinical settings. A significant contributing factor is researchers' inability to create neuroscience-informed paradigms that are easily disseminated, suitable and practical for clinical use.

The goal of the present work is to leverage the strong psychometric properties of a canonical inhibitory control task in the creation of CALM-IT. By targeting a precise pathophysiologically-informed mechanism, and using a platform that is highly engaging and easily disseminated in pediatric samples, the mobile application CALM-IT was designed to bridge the gap between precise mechanisms-driven basic science research and community-based assessment of childhood psychopathology. CALM-IT is designed to leverage the strong methodological design of well-established laboratory based inhibitory control tasks while increasing participant engagement through gamification of the tasks. By using a mobile platform, participant interaction with CALM-IT mirrors those with other mobile-based games with the goal of increasing participant engagement via dynamic stimuli and in-game incentives. The development of CALM-IT would allow clinicians and researchers to screen for and track inhibitory control deficits in the community. This could extend the investigation of inhibitory control beyond the confines of the laboratory setting, therefore facilitating access to a larger and more diverse sample of patients. Ultimately, CALM-IT could lead to novel clinical screening measures for inhibitory control deficits that could be used to inform a personalized course of treatment.

## Study aims

### *Aim 1*: Assessment of feasibility and consistency of CALM-IT

**Aim 1a.**  Our first aim is to assess the feasibility of CALM-IT as a mobile-paradigm. We will assess the ease of dissemination, mobile application engagement, and ability to extract mobile application-based behavioral measures of inhibitory control within a large transdiagnostic pediatric sample.

**Aim 1b.**  Next, we will assess the reliability of mobile application-based behavioral measures of inhibitory control across two sessions of play, spaced one week apart.

### *Aim 2*: Validation of mobile application-based behavioral metrics as measures of inhibitory control

Our second aim is to validate mobile application-based measures of inhibitory control against measures of inhibitory control extracted from four canonical laboratory-based tasks: the Anti-saccade Task [28], the AX Continuous Performance Task [32], the Flanker Task [30], and the Stop Signal Task [31].

### *Aim 3*: **Identify clinical correlates of mobile application-based behavioral metrics**

Finally, our third aim is to identify clinical correlates of mobile application-based measures of inhibitory control. Specifically, we will examine associations between mobile application-based behavioral measures of inhibitory control with shared vs. unique variance of three symptoms present across a wide range of pediatric psychopathologies: ADHD, anxiety, and irritability symptoms.

## Methods

### Participants

Youth aged 8–18 years old will be recruited from the community to participate in research at the National Institute of Mental Health (NIMH). Participants will be recruited as part of a larger protocol that aims to specifically recruit youth with a primary diagnosis of an Anxiety Disorder, Disruptive Mood Dysregulation Disorder (DMDD), ADHD, and youth with no psychiatric diagnosis. Past work within this sample has found that recruiting samples characterized by these clinical diagnoses has resulted in the full range of irritability, anxiety, and ADHD symptoms [51,59–62]. All diagnoses will be assessed by a Board-Certified psychiatrist or licensed psychologist using the Schedule for Affective Disorders and Schizophrenia for School-Age Children-Present and Lifetime version (KSADS-PL; [63]). Exclusionary criteria for participation will include an IQ<70, diagnosis of autism spectrum disorder, past and/or current post-traumatic stress disorder, schizophrenia, or depression, or neurological disorder.

Prior to participation, parents will provide written informed consent and youth will provide written assent. Participants will complete up to three different testing sessions. Administration of CALM-IT will occur outside of the laboratory setting. Families will be sent instructions on how to download CALM-IT onto their mobile device. As such, CALM-IT testing sessions will be completed within a community setting (e.g. at home) at whatever time is most convenient for the family. Parents will be instructed that CALM-IT should be played only by the child with no assistance. Adherence to this procedure will be assessed in a follow up interview. Families will also have the option of coming in for in-laboratory testing where children will complete four canonical inhibitory control tasks (detailed below). Families will receive monetary compensation for completion of CALM-IT. All study procedures have been approved by the NIMH Institutional Review Board.

To investigate the feasibility of CALM-IT (*Aim 1a*), we will recruit 200 youth. For investigation of the stability of CALM-IT (*Aim 1b*), we will invite 75 youth out of the overall sample of 200 to play CALM-IT twice. For validation of mobile application-based behaviors of inhibitory control (*Aim 2*), we will invite 100 youth out of the overall sample of 200 to complete a battery of in-laboratory inhibitory control paradigms. We evaluated each of these proposed sample sizes using the pwr package in R to determine the size of the effect that the sample would allow us to detect using a significance level of 0.05 and a power level of 0.80 [64]. For the feasibility of CALM-IT (*Aim 1a*), our proposed sample size is sufficient to detect effects of $r \geq 0.20$. For the stability of CALM-IT (*Aim 1b*), our proposed sample size is sufficient to detect effects of $r \geq 0.32$. Finally, for the validation of application-based behaviors of inhibitory control (*Aim 2*), the proposed sample size is sufficient to detect effects of $r \geq 0.28$. Thus all proposed samples would allow us sufficient power to detect small to medium effect sizes that are similar to those found within comparable samples [7,47,65,66].

### CALM-IT mobile application

In collaboration with a2 Group, we have developed a mobile application assessing motor inhibition using an engaging, user-friendly game called CALM-IT. Participants are invited to play

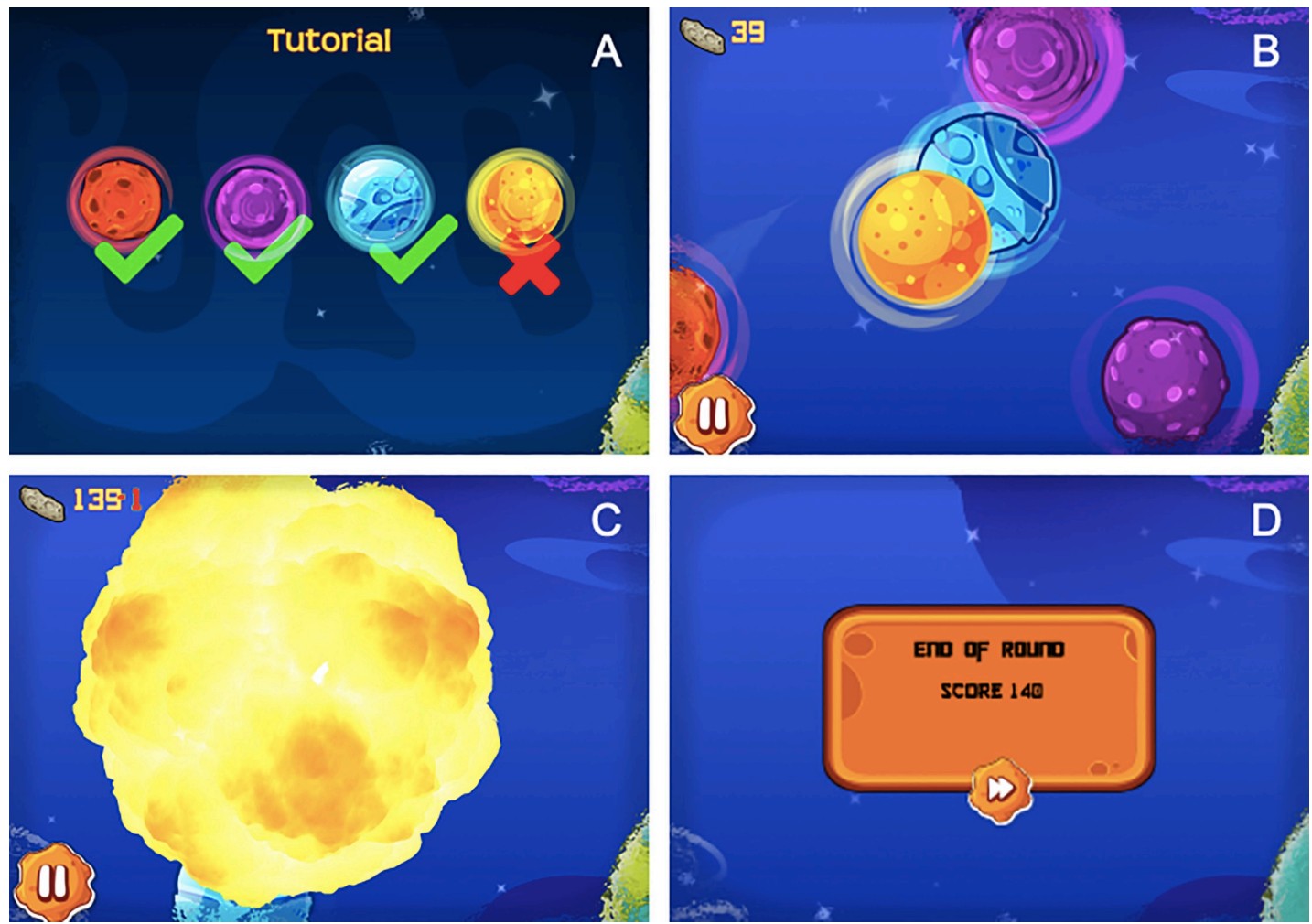

**Fig 1. Screenshots depicting gameplay during CALM-IT.** (A) Tutorial screen with the instructions to swipe targets (red, purple, and blue objects) and avoid swiping stars (yellow) (B) Screen capture from in-level play depicting targets and stars (C) Explosion and negative point indicator feedback in response to incorrectly swiping a star (D) Display of end of level cumulative score.

a game set in space where they are an astronaut exploring different galaxies. Each galaxy represents a distinct level (i.e., block of trials). While exploring the galaxy, participants are instructed via visual cue that that their goal is to destroy space objects by swiping their finger across the screen and hitting each space object, or target. These targets include comets, space rocks, and asteroids that appear in red, purple, or blue. At the same time, participants are instructed to avoid hitting stars which appear in yellow (Fig 1, Panel A). Multiple space objects and stars can appear on the screen at any given time thus requiring participants to swipe at the targets while avoiding the stars (Fig 1, Panel B). If participants hit a star, an explosion animation appears on the screen (Fig 1, Panel C). Participants earn points based on the number of targets they hit (two points per target) and they lose points based on the number of stars they hit (one point per star). A running tally of the score is presented in the top left corner of the screen throughout gameplay and an indicator is displayed when points are lost (Fig 1, Panels B and C). At the end of each level, participants are presented with their cumulative score (Fig 1, Panel D).

CALM-IT is comprised of 10 separate galaxies (e.g. levels). Each galaxy includes 13 stars and 39 targets and takes approximately 1 minute long to complete for a total of approximately 10 minutes of gameplay. Gameplay in the first five galaxies was modeled after the Go/No-Go paradigm [29]; targets represent go-stimuli and stars represent no-go stimuli. The last five galaxies were modeled after the Stop Signal Delay paradigm [31] such that 25% of the targets (e.g. go-stimuli) turn into stars (e.g. no-go stimuli) a a random time within two seconds of appearing on the screen. If a participant hit the go-turned-no-go stimuli, they receive the same explosion feedback image and accompanying sound as one does when you hit a star. Unlike standard Stop Signal Delay paradigms, for the go-turned-no-go stimuli, the time delay between appearing on the screen and turning into the star is fully random and not tied to participant performance.

A subset of participants (n = 75) will complete all 10 levels of the mobile application a second time approximately 1 week after their first time playing. These data will be used to assess the stability of mobile application-derived behavioral measures.

## Mobile application-based behavior

For each galaxy, there are four variables collected corresponding to the presentation of each target and star: (1) whether or not the object was hit, (2) the reaction time of motor response, (3) the duration of time the object was present on the screen, and (4) whether or not the object turns into another object (go-turned-no-go stimuli on galaxies 6–10). All data will be processed separately for galaxies 1–5 and 6–10. Measures of accuracy and reaction time will be calculated for stars and targets separately.

Inhibitory control will be operationalized using a signal detection theory (SDT) approach which can be used to measure sensitivity, or an individual's ability to discriminate signal from noise. Targets represent signal trials whereas stars represent noise trials. Thus, correctly swiping at targets are hit trials, not swiping targets are miss trials, incorrectly swiping at stars are false alarm trials and not swiping stars are correct rejection trials (Fig 2).

## In-laboratory measures of inhibitory control

**Antisaccade task.** The antisaccade task measures motor inhibition in the context of automatic visual saccades towards novel stimuli [28]. Using a mixed event version of the task, participants are instructed to engage in either prosaccade trial, in which they are instructed to direct their eye gaze towards a visual target, or an antisaccade trial, in which they are instructed to direct their eye gaze in the direction *opposite* a visual target. The order of prosaccade and antisaccade trials are fully randomized within each block. Each trial begins with a preparatory period during which participants are presented with either a green or red instructional fixation cross indicating that the next trial will either be a prosaccade or antisaccade trail, respectively. During the prosaccade and antisaccade trials, a yellow visual target is presented for 1 second in a pseudorandomized location 630 pixels or 315 pixels to the left or right of the center of the screen for 1 second. The number of trials with the visual target in each location is equal across prossacade and antisaccade trials. The testing session consists first of a practice block. After completing the practice, participants will complete 3 experimental blocks each with 16 antisaccade and 16 prosaccade trials. The EyeLink 1000 Plus eye tracking system will be used to collect and process eye gaze data (a full description of the eye-tracking set up and eye gaze processing can be found in [57]). For each participant the percentage of correct antisaccade trials will be computed as a measure of successful motor inhibition.

**AX Continuous Performance Task (AXCPT).** The AXCPT is a type of continuous performance task in which children are continuously presented with a series of letters and

**Fig 2. Signal detection theory operationalization of mobile application-based behavior.**

instructed to press a button in response to each presented letter [32]. Children are instructed to press buttons depending on the sequence of letters presented. Specifically, when a child sees a letter pair A followed by X they are instructed to press 2 in response to the A and 3 in response to the X. For all other letter pairs, the child is instructed to press 2 for both letters. Critically, the majority of the time an X appears on the screen, it is proceeded by an A, meaning that pressing 3 in response to the X quickly becomes a learned prepotent response. Trials can be categorized based on the letter pairings, with trials that contain an A cue with an X probe categorized as "AX" trials and trials containing a non-A cue with an X probe categorized as "BX" trials. In these BX trials, participants must inhibit the prepotent response of pressing 3 and instead press 2 in response to the X. Following a practice phase, during which children receive feedback based on their responses, participants will complete three experimental blocks, during which participants complete a total of 150 trials. For each participant, d' context

(% correct AX trials—% incorrect BX trials) will be calculated as a measure of inhibitory control.

**Flanker task.** The flanker task [30] is a well-established task measuring interference effects on cognitive control. In this task, participants are asked to inhibit interfering task irrelevant information in order to engage in the task goal. Participants will be instructed to press the left or right arrow button to indicate the direction of the central arrow in series of five side-by-side arrows centered on the screen. The trial will terminate upon response and participants will be instructed to respond as quickly as they can. Trials are categorized as either congruent or incongruent trials based on the direction of the flanking arrows. Congruent trials correspond to trials where the flanking arrows all point in the same direction as the central arrow. The congruency of the visual stimuli therefore facilitates the correct motor response. In contrast, incongruent trials correspond to trials where the flanking arrows all point in the opposite direction of the central arrow. The incongruency of the visual stimuli therefore interferes with the execution of the correct motor response. Participants first will complete a practice block in which they receive feedback regarding the accuracy of their responses. The experimental task will consist of four blocks, each containing 30 congruent and 30 incongruent trials, presented in randomized order. For each participant, we will extract reaction time metrics for correct responses to congruent and incongruent trials, with the reaction time difference between the two trial types as a measure of inhibitory control efficiency.

**Stop signal task.** During the stop-signal delay task [31], participants are instructed to press the right arrow-key when presented with an X and the left arrow-key when presented with an O. These trials are called "go-trials". On 25% of trials, the X or O will be accompanied by a 1000 Hz auditory "stop cue". When the stop cue is presented, participants are instructed to withhold their motor response to the X or O. These are called "stop-trials". The time elapsed between the presentation of the go-stimulus and the stop cue (stop-signal delay; SSD) varies as a function of performance to calibrate successful inhibition of the go-response to 50% of stop-trials. Initially the SSD is set to 25 0ms and increased by 50 ms following correct responses to stop-trials and decreased by 50 ms following incorrect responses to stop-trials. Following completion of two 16-trial practice blocks, participants will complete 5 experimental blocks with 88 trials each, resulting in a total of 330 go-trials and 110 stop-trials. For each participant, we will extract the stop-signal reaction time (SSRT), the difference between the average SSD and the average reaction time to go-trials, as a measure of inhibitory control.

## Symptom measures

**Irritability.** Irritability symptoms will be measured using the parent- and youth-report versions of the Affective Reactivity Index (ARI, [67]). In addition to item-level responses to the six parent-report and six youth-report items, a total parent-report ARI and child-report ARI score will be calculated as the sum of the six items.

**Anxiety.** Anxiety symptoms will be measured using the parent- and youth-report versions of the Screen for Anxiety Related Emotional Disorders (SCARED; [68]). Subscale scores will be calculated for each of the five subscales (Generalized Anxiety, Panic, School Anxiety, Separation Anxiety, and Social Anxiety) separately for the parent- and youth-report SCARED. Total parent- and child-report SCARED Total scores will also be calculated as the sum across all items.

**ADHD.** ADHD symptoms will be measured using the parent-report Conners Comprehensive Behavior Ratings Scale (CBRS; [69]). ADHD symptoms will be measured using the DSM-IV ADHD Total raw score. Additionally, using results of a confirmatory factor analysis examining the factor loadings of each individual item from the DSM-IV ADHD Total Score

loading on a single latent factor, we have selected the six items with the highest factor loadings (S1 File).

Secondary analyses will be conducted examining associations with the two components of ADHD: hyperactive-impulsivity and inattentiveness. For these analyses, the CBRS DSM-IV Hyperactive-Impulsive and CBRS DSM-IV Inattentive subscales will be used.

**Depression.**    For secondary analyses, depression symptoms will be measured using the parent-report Mood and Feelings Questionnaire (MFQ; [70]). Total scores on the MFQ will be calculated for each participant as a measure of depressive symptoms.

### Bifactor model of symptoms

Latent factor scores for shared vs unique variance of irritability, anxiety, and ADHD symptoms will be estimated using confirmatory factor analyses of an extended bifactor model (Fig 3;

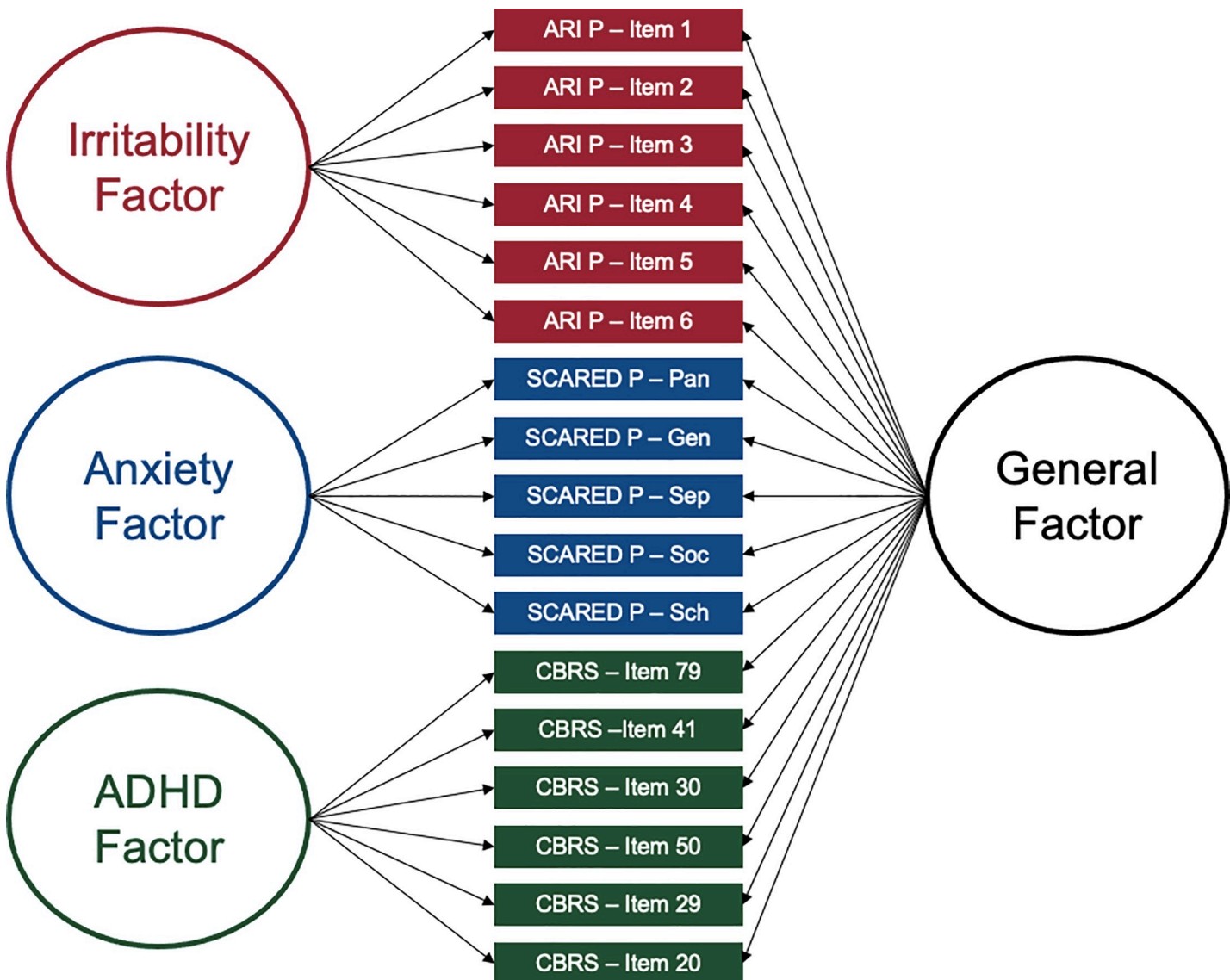

**Fig 3. Bifactor model for clinical symptoms.** Note: ARI P = Parent-Report Affective Reactivity Index, SCARED P = Parent-Report Screen for Anxiety Related Emotional Disorders, CBRS = Conners Comprehensive Behavior Ratings.

[59,61]). A general factor will be estimated using the six parent-report ARI items, five parent-report SCARED subscale scores, and six CBRS items identified by the CFA analysis. An irritability-unique, anxiety-unique, and ADHD-unique factor will be estimated using the ARI items, SCARED subscale scores, and six CBRS items respectively. Consistent with past work, model fit will be assessed using the following criteria: Tucker Lewis Index (TLI) and Comparative Fit Index (CFI) values > .950, and a Root Mean-Square Error of Approximation (RMSEA) value < .08. Factor scores on each of the resulting four latent factors will be extracted for each participant.

## Analytic plan

### Aim 1: Assessment of feasibility and consistency of CALM-IT

**CALM-IT adoption and completion.** First, we will assess mobile application adoption by calculating the percentage of participants who agreed to play the mobile application out of those who were contacted to assess interest. Second, we will assess mobile application completion by calculating the percentage of levels with complete and usable data for each participant who agreed to play the app.

**CALM-It engagement.** Performance on the mobile application will be evaluated to assess whether participants understand the visual instructions and were engaged in the task. The percentage of targets hit and the percentage of stars hit will be assessed as measures of correct engagement and incorrect engagement in task behavior, respectively. Average percent targets hit and percent stars hit will be reported across all participants. Average percent targets hit and percent stars hit will also be examined across each level to assess maintenance of attention and engagement across all levels of the task. Finally, a follow-up interview will be conducted to collect a qualitative assessment of CALM-IT engagement from each participant.

**Extracting mobile application-based measures of inhibitory control.** Using SDT we will extract a measure of sensitivity, or the discrimination between noise and signal, as a measure of inhibitory control using mobile application behavior. The variable d-prime (d') assesses the degree of overlap between the standardized probability of hit trials (i.e. Hit Rate) and the standardized probability of false-alarm trials (i.e. False Alarm Rate; Eq 1; [71,72]). A value of 0 represents an inability to discriminate signal from noise with increasing values representing being better able to discriminate signal from noise. Values for d' will be calculated for each participant using the psycho package in *R* [73].

$$d' = z(Hit\ Rate) - z(False\ Alarm\ Rate) \tag{1}$$

Using d' we will assess both individual differences in inhibitory control and average discrimination on the task. An average d' value will be assessed across all participants to whether, on average, participants are able to discriminate between noise and signal on the task. Both the average d' value and the receiver operating characteristic (ROC) curve will be reported to provide a graphical visualization of how often false alarms versus hits occur at any level of sensitivity.

**Stability of mobile application-based behaviors.** In the subset of participants who completed the mobile application twice, one week apart from another, the temporal stability of mobile application behaviors will be assessed using test-retest reliability. Specifically, we will examine test-retest reliability through examination of correlation between time-one with time-two measures of percentage of targets hit, percentage of stars hit, and d'. Consistency will also be reported using the intraclass correlation coefficient (ICC). In contrast to the Pearson's correlation coefficient, ICC captures not only the degree of correlation, but also the agreement across measurements [74,75]. Because we are investigating test-retest reliability, or the stability

across measurement, we will use ICC(3,1) which is a two-way mixed effects assessment of consistency within a single measurement. An ICC value of less than 0.40 would indicate poor consistency, an ICC value of 0.40–0.70 would indicate moderate consistency, an ICC value of 0.70–0.90 would indicate good consistency, and an ICC value greater than 0.90 would indicate excellent consistency.

### *Aim 2*: Validation of mobile application-based behavioral metrics as measures of inhibitory control

In order to robustly measure inhibitory control, we will employ a framework through which a latent factor of inhibitory control is computationally derived from measurements of motor inhibition across four canonical inhibitory control laboratory-based tasks described above [59]. This latent factor will be computed using confirmatory factor analysis with a measure of inhibitory control from each of the four tasks loading onto a single latent factor. By leveraging latent variable analysis across a range of tasks that differ in their behavioral goals, response modality, and context, we can quantify a relatively pure measure of inhibitory control that is immune from task-specific impurities and thus increase statistical power. Next, we will extract individual-level inhibitory control latent factor scores for each participant to examine associations with mobile application-based measures of inhibitory control. We will conduct a multiple regression analysis with CALM-IT d' as a predictor of inhibitory control latent factor scores, with age, sex, order of task completion (e.g., CALM-IT or laboratory testing administered first), and IQ entered as covariates.

### *Aim 3*: Identify clinical correlates of mobile application-based behavioral metrics

First, raw bivariate correlations between raw total scores on the five symptom measurements and CALM-IT d' will be examined. Next, associations with shared vs. unique variances of anxiety, irritability, and ADHD symptoms will be examined using multiple regression analyses. All four latent factor scores from the bifactor model will be entered as concurrent predictors of CALM-IT d'. For all models, any demographic variable that varies as a function of either symptoms and/or CALM-IT d' will be included as a covariate (i.e., age, sex, and IQ). Finally, secondary analyses will be run examining bivariate correlations between measures of hyperactive-impulsivity, inattentiveness, and depression.

## Summary

The present protocol introduces a novel, user-friendly mobile application aimed to assess inhibitory control, a precise and well-defined construct that has been posited as one critical mechanism underlying psychopathology in youth. There is a profound need for mental health treatment research efforts to use neuroscience-based assessments to inform treatment decisions. However, such assessments are not accessible to community providers. The goal of this work is to bridge that gap by creating and facilitating the usage of an accessible and engaging novel mobile application that probes a validated behavioral construct. The present protocol would lay the groundwork for an important line of future work that could provide researchers and clinicians a multifaceted tool to measure multiple aspects of inhibitory control. For example, future versions of the app could include manipulation of the stimuli presented such that participants are required to update the which stimuli represents the stop-stimuli, thus targeting one's ability to flexibly deploy inhibitory control. Furthermore, an adaptive version of CALM-IT that increases difficulty based on participants' individual performance could function as

an intervention aimed at improving inhibitory control. Validation of this neuroscience-informed mobile application represents a critical first step forward in bridging the gap between precise, mechanism-driven basic science research and community-based assessment and treatment of childhood psychopathology. CALM-IT is poised to provide an accessible tool for clinicians in the community to make clinical and treatment decisions based on neuroscience-based mechanisms.

## Supporting information

**S1 File. Identification of ADHD items for inclusion in bifactor model of symptoms.** Description of methods and results of a confirmatory factor analysis of the 18 items measuring ADHD symptoms from the Conners Comprehensive Behavior Ratings Scale. (PDF)

## Author Contributions

**Conceptualization:** Elise M. Cardinale, Reut Naim, Simone P. Haller, Melissa A. Brotman.

**Methodology:** Elise M. Cardinale, Reut Naim, Simone P. Haller, Christian Botz-Zapp, Jessica Bezek, David C. Jangraw, Melissa A. Brotman.

**Writing – original draft:** Ramaris German, Melissa A. Brotman.

**Writing – original sdraft:** Elise M. Cardinale.

**Writing – review & editing:** Reut Naim, Simone P. Haller, Ramaris German, Christian Botz-Zapp, Jessica Bezek, David C. Jangraw, Melissa A. Brotman.

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
