## [Decision Letter · Decision Letter 0]

3 Mar 2021

PONE-D-20-39885

Rationale and validation of a novel mobile application probing motor inhibition: Proof of concept of CALM-IT

PLOS ONE

Dear Dr. Cardinale,

Thank you for submitting your manuscript to PLOS ONE. After careful consideration, we feel that it has merit but does not fully meet PLOS ONE’s publication criteria as it currently stands. Therefore, we invite you to submit a revised version of the manuscript that addresses the points raised during the review process.

We look forward to receiving your revised manuscript.

Kind regards,

Veena Kumari

Academic Editor

PLOS ONE

Journal Requirements:

2. Please note that PLOS ONE has specific guidelines on software sharing (http://journals.plos.org/plosone/s/materials-and-software-sharing#loc-sharing-software) for manuscripts whose main purpose is the description of a new software or software package. In this case, new software must conform to the Open Source Definition (https://opensource.org/docs/osd) and be deposited in an open software archive. Please see http://journals.plos.org/plosone/s/materials-and-software-sharing#loc-depositing-software for more information on depositing your software.

4.We noticed you have some minor occurrence of overlapping text with the following previous publication(s), which needs to be addressed:

https://www.cambridge.org/core/journals/development-and-psychopathology/article/abs/inhibitory-control-and-emotion-dysregulation-a-framework-for-research-on-anxiety/3DB40FE1CD1D6293778A9BC0272F3005

In your revision ensure you cite all your sources (including your own works), and quote or rephrase any duplicated text outside the methods section. Further consideration is dependent on these concerns being addressed.

Reviewers' comments:

Reviewer's Responses to Questions

**Comments to the Author**

1. Does the manuscript provide a valid rationale for the proposed study, with clearly identified and justified research questions?

Reviewer #1: Partly

2. Is the protocol technically sound and planned in a manner that will lead to a meaningful outcome and allow testing the stated hypotheses?

Reviewer #1: Partly

3. Is the methodology feasible and described in sufficient detail to allow the work to be replicable?

Reviewer #1: Yes

4. Have the authors described where all data underlying the findings will be made available when the study is complete?

Reviewer #1: No

5. Is the manuscript presented in an intelligible fashion and written in standard English?

Reviewer #1: Yes

6. Review Comments to the Author

You may also provide optional suggestions and comments to authors that they might find helpful in planning their study.

Reviewer #1: This manuscript is a protocol for a validation study of a mobile application for inhibitory control evaluation in young people. It presents an interesting idea of measuring inhibitory control via a mobile application which presumably is a more feasible way in comparison to other experimental approaches when used in the community settings. The protocol summarises the main details of the proposed validation study. Several important details in the methods are missing which should be clarified. I recommend this protocol for publication after major revision.

Abstract:

The abstract could be more specific and include the importance of this research and the possibility of future application of the CALM-IT in clinical practice.

Introduction:

Page 4: “However, these tasks tend to be time-intensive, repetitive, and expensive, making them infeasible in a community setting”. Can the authors add an example of the tasks which are not feasible to conduct in a community setting and why? In what sense is the new application CALM-IT less repetitive? My understanding is that it comes from well-established paradigms (Go/No-Go and Stop-Signal Task) which are repetitive but their use in the community is more about the use of appropriate device rather than creating a whole new paradigm or measuring instrument.

Page 5: “While literature examining treatment of youth psychopathology using interventions specifically targeting inhibitory control is limited, previous work shows a reduction of mood symptoms following simulant medication treatment for co-occurring attentional (Posner et al., 2014; Towbin et al., 2019).”. What is exactly meant by youth psychopathology? Diagnoses of mental illness or behavioural problems (e.g., at school, with peers, etc.)? Could you please provide examples of specific psychopathology conditions in youth treated in inhibitory control and how are these findings connected with the present study?

Methods: This part is described into considerable details, however, to make the methodology replicable several details are missing.

Participants: The protocol does not clearly state the age range of participants the authors plan to recruit for their study and the process of their recruitment and selection (randomly approached participants, all participants from a certain clinic will be offered the pre-screening and participation, etc.). How will the authors assure that the participants will represent a whole range of symptoms? Moreover, if the participants will be under-aged, I would recommend clarifying the role of parents or legal guardians in the process. Whether they will be present or not during the use of the app and additional evaluations and how the authors plan for potential interference from the parents in the app testing (e.g., using the app instead of the participant recruited). It is unclear if the app will be tested/used at home or in a controlled environment (e.g., a laboratory or a clinic).

What is the rationale behind selecting the particular diagnoses of anxiety (but not depression), ADHD (including ADD?), DMDD? The topic of inhibitory control problems in children with these selected diagnoses could be more elaborated also in the introduction. To summarise the existing evidence of this problem may give a better understanding of the importance of this study and the CALM-IT application.

Stimuli: The application seems to present a limited number of trials and stimuli types. Did the authors consider using different colours and shapes of stimuli for each application testing session to avoid habituation for specific stimuli? For example, the no-go stimuli are always yellow stars, changing the colour or type of the no-go stimulus for each level may reflect more on the flexibility of each participant to inhibit inappropriate response in a variety of different scenarios.

Effect size calculation: What is the rationale behind selecting the effects of r between .20 and .32. Could the authors give examples of similar studies or otherwise justify the selected effects and/or the effect size calculation?

Symptom measures: I would recommend to the authors to consider measuring the impulsivity of participants also by using standardised questionnaires to investigate the different facets of impulsive behaviour. Some established measures as UPPS-P can be used in children.

Assessing the engagement: This seems to be assessed only based on the performance metrics from the app. I would recommend to the authors considering running a focus group or using a questionnaire of user experience to gain more information on the engagement and the usability of the app. This can be an important source of information for adjusting apps in development based on real user-experience feedback. If the app engagement is assessed only based on the performance metrics, how will the authors distinguish between poor performance in the app based on boredom or lack of engagement versus a poor performance caused by very poor inhibitory control abilities?

7. PLOS authors have the option to publish the peer review history of their article (what does this mean?). If published, this will include your full peer review and any attached files.

Reviewer #1: **Yes: **Martina Vanova

---

## [Author Response · Author response to Decision Letter 0]

16 Apr 2021

Editor’s Comments:

1.Please ensure that your manuscript meets PLOS ONE's style requirements, including those for file naming. The PLOS ONE style templates can be found at https://journals.plos.org/plosone/s/file?id=wjVg/PLOSOne_formatting_sample_main_body.pdf and https://journals.plos.org/plosone/s/file?id=ba62/PLOSOne_formatting_sample_title_authors_affiliations.pdf

We have reviewed the PLOS ONE style templates and ensured that our manuscript meets the PLOS ONE’s style requirements.

2.Please note that PLOS ONE has specific guidelines on software sharing (http://journals.plos.org/plosone/s/materials-and-software-sharing#loc-sharing-software) for manuscripts whose main purpose is the description of a new software or software package. In this case, new software must conform to the Open Source Definition (https://opensource.org/docs/osd) and be deposited in an open software archive. Please see http://journals.plos.org/plosone/s/materials-and-software-sharing#loc-depositing-software for more information on depositing your software.

CALM-IT is a mobile behavioral task and not specifically a software or software package. We would happily share access to the mobile application and once we have completed this pilot and app-development is completed. 

We now include captions at the end of our manuscript for the supplementary materials. 

4. We noticed you have some minor occurrence of overlapping text with the following previous publication(s), which needs to be addressed: https://www.cambridge.org/core/journals/development-and-psychopathology/article/abs/inhibitory-control-and-emotion-dysregulation-a-framework-for-research-on-anxiety/3DB40FE1CD1D6293778A9BC0272F3005 In your revision ensure you cite all your sources (including your own works), and quote or rephrase any duplicated text outside the methods section. Further consideration is dependent on these concerns being addressed.

We apologize for any overlapping text in the submitted manuscript with our prior work. After reviewing the two manuscripts closely it appears that the overlapping text is in the description of the antisaccade and flanker in-laboratory tasks. We have revised the language to no longer overlap while maintaining accuracy and clarity:

(p. 12): “Each trial begins with a preparatory period during which participants are presented with either a green or red instructional fixation cross indicating that the next trial will either be a prosaccade or antisaccade trail, respectively. During the prosaccade and antisaccade trials, a yellow visual target is presented for 1 second in a pseudorandomized location 630 pixels or 315 pixels to the left or right of the center of the screen for 1 second. The number of trials with the visual target in each location is equal across prossacade and antisaccade trials. The testing session consists first of a practice block. After completing the practice, participants will complete 3 experimental blocks each with 16 antisaccade and 16 prosaccade trials. The EyeLink 1000 Plus eye tracking system will be used to collect and process eye gaze data (a full description of the eye-tracking set up and eye gaze processing can be found in Cardinale et al., 2019).”

(p. 13): “Participants will be instructed to press the left or right arrow button to indicate the direction of the central arrow in series of five side-by-side arrows centered on the screen. The trial will terminate upon response and participants will be instructed to respond as quickly as they can. Trials are categorized as either congruent or incongruent trials based on the direction of the flanking arrows. Congruent trials correspond to trials where the flanking arrows all point in the same direction as the central arrow. The congruency of the visual stimuli therefore facilitates the correct motor response. In contrast, incongruent trials correspond to trials where the flanking arrows all point in the opposite direction of the central arrow. The incongruency of the visual stimuli therefore interferes with the execution of the correct motor response.”

Reviewer #1 :

This manuscript is a protocol for a validation study of a mobile application for inhibitory control evaluation in young people. It presents an interesting idea of measuring inhibitory control via a mobile application which presumably is a more feasible way in comparison to other experimental approaches when used in the community settings. The protocol summarises the main details of the proposed validation study. Several important details in the methods are missing which should be clarified. I recommend this protocol for publication after major revision.

1. The abstract could be more specific and include the importance of this research and the possibility of future application of the CALM-IT in clinical practice.

We have expanded the abstract to now include more specific language and a description of the importance of this research to future application in clinical practice (p. 2):

“The development of CALM-IT has significant implications for the ability to screen for inhibitory control deficits in the community by both clinicians and researchers. By facilitating assessment of inhibitory control outside of the laboratory setting, researchers could have access to larger and more diverse samples. Additionally, in the clinical setting, CALM-IT represents a novel clinical screening measure that could be used to determine personalized courses of treatment based on the presence of inhibitory control deficits.”

2. Page 4: “However, these tasks tend to be time-intensive, repetitive, and expensive, making them infeasible in a community setting”. Can the authors add an example of the tasks which are not feasible to conduct in a community setting and why? In what sense is the new application CALM-IT less repetitive? My understanding is that it comes from well-established paradigms (Go/No-Go and Stop-Signal Task) which are repetitive but their use in the community is more about the use of appropriate device rather than creating a whole new paradigm or measuring instrument.

We appreciate the opportunity to provide more clarity to this statement. We have revised the above referenced sentence (p. 4):

“However, these tasks tend to be time intensive, repetitive, and expensive, making them difficult to effectively deploy within a community setting.”

Additionally, we believe that there are two critical aspects that need to be more effectively communicated in our introduction. First, we now provide an example of the types of demands that traditional in-laboratory tasks have that make it difficult to translate to the community setting (p. 4):

“For example, some tasks rely on eye-tracking technology (Hallett, 1978) that requires equipment and specific environmental controls (i.e., the luminance in the room and participant head position relative to the presented stimuli) while others involve large numbers of repetitive trials presenting single simplistic stimuli one at a time (i.e., letters or shapes) that require long periods of sustained attention (Donders, 1969; Eriksen, 1995; Logan & Cowan, 1984; Rutschmann, 1977).”

Second, we now more explicitly discuss the aspects of CALM-IT that allow it to be both more easily accessible and engaging while still leveraging the methodological designs from well-established paradigms (p. 6-7):

“CALM-IT is designed to leverage the strong methodological design of well-established laboratory based inhibitory control tasks while increasing participant engagement through gamification of the tasks. By using a mobile platform, participant interaction with CALM-IT mirrors those with other mobile-based games with the goal of increasing participant engagement via dynamic stimuli and in-game incentives.”

3. “While literature examining treatment of youth psychopathology using interventions specifically targeting inhibitory control is limited, previous work shows a reduction of mood symptoms following simulant medication treatment for co-occurring attentional (Posner et al., 2014; Towbin et al., 2019).”. What is exactly meant by youth psychopathology? Diagnoses of mental illness or behavioural problems (e.g., at school, with peers, etc.)? Could you please provide examples of specific psychopathology conditions in youth treated in inhibitory control and how are these findings connected with the present study?

We apologize for the omission. We now specifically state the youth psychopathology referenced in the cited work (p. 5):

“While literature examining treatment of youth psychopathology, such as attention deficit hyperactivity disorder (ADHD) and irritability, using interventions specifically targeting inhibitory control is limited, previous work shows a reduction of mood symptoms, including anxiety, following simulant medication treatment for co-occurring attentional (Posner et al., 2014; Towbin et al., 2019). These findings provide some evidence suggesting that inhibitory control may reflect a key behavioral mechanism underlying the emergence of mood symptoms and treatment response.”

4. The protocol does not clearly state the age range of participants the authors plan to recruit for their study and the process of their recruitment and selection (randomly approached participants, all participants from a certain clinic will be offered the pre-screening and participation, etc.). How will the authors assure that the participants will represent a whole range of symptoms? Moreover, if the participants will be under-aged, I would recommend clarifying the role of parents or legal guardians in the process. Whether they will be present or not during the use of the app and additional evaluations and how the authors plan for potential interference from the parents in the app testing (e.g., using the app instead of the participant recruited). It is unclear if the app will be tested/used at home or in a controlled environment (e.g., a laboratory or a clinic).

The participants section has been updated to provide more information regarding recruitment procedures and the targeted age range and clinical psychopathology (p. 8):

“Youth aged 8 – 18 years old will be recruited from the community to participate in research at the National Institute of Mental Health (NIMH). Participants will be recruited as part of a larger protocol that aims to specifically recruit youth with a primary diagnosis of an Anxiety Disorder, Disruptive Mood Dysregulation Disorder (DMDD), ADHD, and youth with no psychiatric diagnosis. Past work within this sample has found that recruiting samples characterized by these clinical diagnoses has resulted in the full range of irritability, anxiety, and ADHD symptoms (Cardinale, Kircanski, et al., 2019; Kircanski et al., 2017, 2018; Stoddard et al., 2017; Tseng et al., 2018).”

We also now include a discussion of the role of parents or legal guardians and the testing environment in the discussion of our procedures (p. 9):

“Prior to participation, parents will provide written informed consent and youth will provide written assent. Participants will complete up to three different testing sessions. Administration of CALM-IT will occur outside of the laboratory setting. Families will be sent instructions on how to download CALM-IT onto their mobile device. As such, CALM-IT testing sessions will be completed within a community setting (e.g. at home) at whatever time is most convenient for the family. Parents will be instructed that CALM-IT should be played only by the child with no assistance. Adherence to this procedure will be assessed in a follow up interview. Families will also have the option of coming in for in-laboratory testing where children will complete four canonical inhibitory control tasks (detailed below). Families will receive monetary compensation for completion of CALM-IT. All study procedures have been approved by the NIMH Institutional Review Board.”

5. What is the rationale behind selecting the particular diagnoses of anxiety (but not depression), ADHD (including ADD?), DMDD? The topic of inhibitory control problems in children with these selected diagnoses could be more elaborated also in the introduction. To summarise the existing evidence of this problem may give a better understanding of the importance of this study and the CALM-IT application.

The reviewer raises an important point about the relevance of depression. Epidemiological studies demonstrate that youth with anxiety and DMDD often develop depression (Brotman et al., 2006; Copeland et al., 2014; Eyre et al., 2019; Pine et al., 1998). As such, depression may be a relevant construct within the proposed sample. We now include a secondary analysis of depression using the Mood and Feelings Questionnaire:

(p. 15): “For secondary analyses, depression symptoms will be measured using the parent-report Mood and Feelings Questionnaire (MFQ; Ancold et al., 1995). Total scores on the MFQ will be calculated for each participant as a measure of depressive symptoms.”

(p. 19): “Finally, secondary analyses will be run examining bivariate correlations between measures of hyperactive-impulsivity, inattentiveness, and depression.”

We now also include additional discussion of the prevalence and comorbidity of anxiety, ADHD, and DMDD in childhood and adolescence and associations with inhibitory control in the introduction (p. 4-5):

“The proposed work could be of particular importance in relation to anxiety, irritability, and ADHD symptoms as these symptoms are common, co-occurring symptoms that are present across a wide range of childhood psychopathology (Brotman et al., 2017; Karalunas et al., 2019; Leibenluft, 2017; Pine, 2007), and are implicated in the later development of mood disorders including depression (Brotman et al., 2006; Copeland et al., 2014; Eyre et al., 2019, 2019; Pine et al., 1998, p. 199). Furthermore, aberrant cognitive control more broadly has been implicated as potential mechanism associated with the presentation of anxiety (Cardinale, Subar, et al., 2019; Eysenck et al., 2007; Moser et al., 2013), irritability (Chaarani et al., 2020; Deveney et al., 2018; Tseng et al., 2018), and ADHD (Durston et al., 2002; Willcutt et al., 2005) symptoms.”

6. Stimuli: The application seems to present a limited number of trials and stimuli types. Did the authors consider using different colours and shapes of stimuli for each application testing session to avoid habituation for specific stimuli? For example, the no-go stimuli are always yellow stars, changing the colour or type of the no-go stimulus for each level may reflect more on the flexibility of each participant to inhibit inappropriate response in a variety of different scenarios.

The reviewer brings up a very interesting recommendation regarding the study of the flexibility of inhibitory control in the context of changing task demands/stimuli characteristics. We believe that the first critical step is to first establish validity of CALM-IT as a measure of baseline inhibitory control and thus have limited versions of the game to two, one modeled after go-no-go and one modeled after stop signal delay paradigms. This also allows us to keep the game play to a reasonable length of time while maximizing the amount of data collected. 

However, the reviewer’s suggestion would be a very important avenue of future work as well as potentially other manipulations of the stimuli that could be leveraged to train inhibitory control. While this is out of the scope of the current proposed project, we now include additional discussion of potential future versions of CALM-IT should the proposed study confirm valid measurement of inhibitory control using the current proposed version of CALM-IT (p. 19):

“The present protocol would lay the groundwork for an important line of future work that could provide researchers and clinicians a multifaceted tool to measure multiple aspects of inhibitory control. For example, future versions of the app could include manipulation of the stimuli presented such that participants are required to update the which stimuli represents the stop-stimuli, thus targeting one’s ability to flexibly deploy inhibitory control. Furthermore, an adaptive version of CALM-IT that increases difficulty based on participants’ individual performance could function as an intervention aimed at improving inhibitory control.”

7. Effect size calculation: What is the rationale behind selecting the effects of r between .20 and .32. Could the authors give examples of similar studies or otherwise justify the selected effects and/or the effect size calculation?

We realize that the language used in the description of the power analyses suggests that we began with a priori effect sizes of r between .20 and .32 however, those effects were reported as a function of the sample size to provide a reader with context or the size of effects that could be detected with a power of .80 at a significance level of alpha = .05. We have revised the language in this section to provide more clarity (p. 9-10):

“To investigate the feasibility of CALM-IT (Aim 1a), we will recruit 200 youth. For investigation of the stability of CALM-IT (Aim 1b), we will invite 75 youth out of the overall sample of 200 to play CALM-IT twice. For validation of mobile application-based behaviors of inhibitory control (Aim 2), we will invite 100 youth out of the overall sample of 200 to complete a battery of in-laboratory inhibitory control paradigms. We evaluated each of these proposed sample sizes using the pwr package in R to determine the size of the effect that the sample would allow us to detect using a significance level of 0.05 and a power level of 0.80 (Stephane, 2020). For the feasibility of CALM-IT (Aim 1a), our proposed sample size is sufficient to detect effects of r≥0.20. For the stability of CALM-IT (Aim 1b), our proposed sample size is sufficient to detect effects of r≥0.32. Finally, for the validation of application-based behaviors of inhibitory control (Aim 2), the proposed sample size is sufficient to detect effects of r≥0.28. Thus all proposed samples would allow us sufficient power to detect small to medium effect sizes that are similar to those found within comparable samples (Bari & Robbins, 2013; Cardinale, Subar, et al., 2019; Derakshan et al., 2009; Haller et al., 2020).”

8. Symptom measures: I would recommend to the authors to consider measuring the impulsivity of participants also by using standardised questionnaires to investigate the different facets of impulsive behaviour. Some established measures as UPPS-P can be used in children.

Unfortunately, due to demands placed on our pediatric patients with severe clinical impairment within the current protocol, we are hesitant to add additional measures. However, we appreciate the important point made by the reviewer regarding impulsivity specifically and have added secondary analyses in which we will specifically examine associations with the two components of ADHD (impulsive hyperactivity and inattentiveness) as measured by the Conners Comprehensive Behavior Ratings Scale. Moreover, we will look specifically at the clinician endorsed items for on the ADHD module of the Schedule for Affective Disorders and Schizophrenia for School-Age Children-Present and Lifetime version.

(p. 15): “Secondary analyses will be conducted examining associations with the two components of ADHD: hyperactive-impulsivity and inattentiveness. For these analyses, the CBRS DSM-IV Hyperactive-Impulsive and CBRS DSM-IV Inattentive subscales will be used.”

(p. 19): “Finally, secondary analyses will be run examining bivariate correlations between measures of hyperactive-impulsivity, inattentiveness, and depression.”

9. Assessing the engagement: This seems to be assessed only based on the performance metrics from the app. I would recommend to the authors considering running a focus group or using a questionnaire of user experience to gain more information on the engagement and the usability of the app. This can be an important source of information for adjusting apps in development based on real user-experience feedback. If the app engagement is assessed only based on the performance metrics, how will the authors distinguish between poor performance in the app based on boredom or lack of engagement versus a poor performance caused by very poor inhibitory control abilities?

The degree to which participants engage in experimental tasks is a persistent issue across measurement and construct. Unfortunately, this issue is present even for the most canonical in-laboratory measures of inhibitory control. One of central aims of CALM-IT is to create a more engaging measurement of inhibitory control that could limit confounding issues of loss of attention or limited motivation. Thus, we agree with the reviewer that it is critical that we evaluate the degree to which CALM-IT was in fact engaging. To improve our evaluation of engagement we now additional propose to examine our measure of app engagement across levels (p. 16):

“Average percent targets hit and percent stars hit will also be examined across each level to assess maintenance of attention and engagement across all levels of the task.”

Additionally, we will now include a follow up interview as part of our procedures to provide feedback from participants regarding their experience playing the game and specifically the degree to which they report being engaged (p. 16):

“Finally, a follow-up interview will be conducted to collect a qualitative assessment of CALM-IT engagement from each.”

---

## [Editor Report · Decision Letter 1]

12 May 2021

Rationale and validation of a novel mobile application probing motor inhibition: Proof of concept of CALM-IT

PONE-D-20-39885R1

Dear Dr. Cardinale,

We’re pleased to inform you that your manuscript has been judged scientifically suitable for publication and will be formally accepted for publication once it meets all outstanding technical requirements.

Kind regards,

Veena Kumari

Academic Editor

PLOS ONE
---

## [Editor Report · Acceptance letter]

25 May 2021

PONE-D-20-39885R1 

Rationale and validation of a novel mobile application probing motor inhibition: Proof of concept of CALM-IT 

Dear Dr. Cardinale:

I'm pleased to inform you that your manuscript has been deemed suitable for publication in PLOS ONE. Congratulations! Your manuscript is now with our production department. 

Kind regards, 

on behalf of

Dr. Veena Kumari 

Academic Editor

PLOS ONE